# Private practitioners' practices for tuberculosis management in a city largely served by the private health sector in Uganda

Judith Amutuhaire Ssemasaazi[1,2]*, Felix Bongomin[3], Rebecca Akunzirwe[1], Joan Rokani Bayowa[1], Emmanuel Ssendikwanawa[1], Cherop Adolphus[1], Ronald Muganga Kivumbi[1], Joan N. Kalyango[1,4], Ezekiel Mupere[1,5], Phiona Ekyaruhanga[5,6], Achilles Katamba[1,2,6]

1 Clinical Epidemiology Unit, College of Health Sciences, Makerere University, Kampala, Uganda,
2 Department of Medicine, College of Health Sciences, Makerere University, Kampala, Uganda, 3 Faculty of Medicine, Department of Medical Microbiology and Immunology, Gulu University, Gulu, Uganda,
4 Department of Pharmacy, College of Health Sciences, Makerere University, Kampala, Uganda,
5 Department of Pediatrics, College of Health Sciences, Makerere University, Kampala, Uganda, 6 Makerere University Lung Institute, Kampala, Uganda

* amutuhairejudith@gmail.com

**Data Availability Statement:** The de-identified minimal data set available at https://github.com/jamutuhaire/Minimal-data-set_JudithAS/tree/main.

## Abstract

### Background

Globally, tuberculosis (TB) remains a significant cause of morbidity and mortality having caused 1.6 million deaths in 2021. Uganda is a high TB burden country with a large private sector that serves close to 60% of the urban population. However, private for-profit health facilities' involvement with the National TB and Leprosy Program (NTLP) activities remains poor. This study evaluated the practices of diagnosis and treatment of pulmonary tuberculosis (PTB) and associated factors among practitioners in private for-profit (PFP) healthcare facilities in Kampala, Uganda.

### Methods

We conducted a cross-sectional study among randomly selected private practitioners in Uganda's largest city, Kampala. A structured questionnaire was used for data collection. Descriptive statistics and generalized linear models with log Poisson link were used to analyze data. Practices were graded as standard or substandard.

### Results

Of the 630 private practitioners studied, 46.2% (95% confidence interval (CI): 26.6 to 67.1) had overall standard practices. Being a laboratory technician (prevalence ratio (PR) = 2.7, p< 0.001) or doctor (PR = 1.2, p< 0.001), a bachelor's degree level of qualification (PR = 1.1, p = 0.021), quarterly supervision by the national TB program (PR = 1.3, p = 0.023), and acceptable knowledge of the practitioner about TB (PR = 1.8, p<0.001) were significantly associated with standard practices.

**Funding:** This original research was funded by a student research scholarship awarded to Judith Amutuhaire-Ssemasaazi by the Africa Center of Excellence in Materials, Product Development and Nanotechnology (MAPRONANO ACE) Makerere University. Details about this student support fund can be accessed at https://mapronano.mak.ac.ug. The funders had no role in the study design, data collection, analysis, the decision to publish, or the preparation of the manuscript. There was no additional external funding received for this study.

**Competing interests:** The authors have declared that no competing interests exist.

**Abbreviations:** DOTS, Directly Observed Therapy Short Course; HIV, Human Immuno-Deficiency Virus; HSSIP, Health Sector Strategic and Investment Plan; ISTC, International Standards of Tuberculosis Care; M.O.H, Ministry of Health; NGOs, Non-Governmental Organizations; NTLP, National Tuberculosis and Leprosy program; PFPs, Private for-profit health facilities; PHPs, Private Health Practitioners; PHR, Partners for Health Reform; PNFPs, Private Not for Profit Organizations; PPPH, Public-Private Partnership for Health; PPs, Private Practitioners; SPARK-TB, Slum Partnerships to Respond to Tuberculosis in Kampala; TB, Tuberculosis; TCMPs, Traditional and Complementary Medicine Practitioners; WHO, World Health Organization.

## Conclusions

The practices of TB management for practitioners from the PFP facilities in Kampala are suboptimal and this poses a challenge for the fight against TB given that these practitioners are a major source of primary health care in the city.

## Background

Tuberculosis (TB) is one of the top 10 causes of death worldwide, and until the COVID-19 pandemic, it was a leading cause of death from a single infectious agent ranking above HIV/AIDS [1]. An estimated 10.6 million people developed TB and 1.6 million TB deaths were estimated globally in 2021 [2]. Uganda is one of the world's high TB burden countries with an incidence of 234 and a prevalence of 253 per 100000 population [3]. However, not every person that suffers TB is identified, enrolled into care, and notified to authorities [4].

In 2021 there were 4.2 million missing cases of TB and previous data has shown that countries with large private sectors contribute the largest to the proportion of missing TB cases [4]. Uganda's private sector accounts for 60% to 70% of the frontline health services [5] and from a previous study, 21.4% of patients with TB symptoms seek care from a private facility [6]. The 2030 targets of 90% reduction in TB deaths and 80% reduction in TB incidence [7] will require strengthened and expanded private provider engagement in TB care [8].

The National Tuberculosis and Leprosy Program(NTLP) rarely collaborates with private for profit health facilities (PFPs) [6] with only 96 PFPs involved country wide. In 2001, only one-third of the private clinics in Kampala practiced a minimal standard of appropriate TB care [9]. In 2009, a situational analysis of Kampala directly observed therapy short course (DOTS) programme showed that private health facilities had limited ability to manage TB with 95% of health workers in PFPs lacking the knowledge and skills to manage TB [10]. Studies have shown that patients who seek care from PFPs incur delays in the TB diagnostic pathway [11,12]. Elsewhere, knowledge of TB management, involvement in national TB programs and continuous training with these programs and qualifications of practitioners have been found to be associated with standards of TB diagnosis and treatment among private practitioners [9,12–15]. In Uganda, the gains of a public private partnership (PPP) were well demonstrated between 2008–2011 where such a collaboration increased referrals from presumptive multidrug-resistant tuberculosis cases by >10-fold improving access to timely drug susceptibility testing [16]. The private health sector in Uganda comprises a significant portion of health facilities with 2975 private for profit facilities out of the 6397 health facilities in the country [17]. However, no recent study has assessed practitioners in PFPs on their practices of TB care. This study therefore aimed to assess practices of diagnosis and treatment of pulmonary TB (PTB) among private practitioners in Kampala.

## Methods

### Study design

We conducted a cross-sectional study between March and July 2020.

### Study setting

The study was conducted in private for-profit clinics and hospitals in Uganda's capital city, Kampala. Kampala was estimated to have a population of about 1,507,080 people at night but this reaches almost 4 million during the day as per the population census in 2014. According to the Kampala Capital City Authority (KCCA) health facility census, Kampala district in

particular has 1442 registered health facilities of which 1367(94%) facilities are private for profit, 1033(76%) have laboratory services and 881(84%) were offering diagnostic and treatment services for TB and HIV services at the time of the census [18]. In the NTLP annual report 2017/2018, Kampala contributed the highest portion of incident TB cases in the country at 29.1% (14676 of 50390 cases) compared to other districts in the country.

## Study population

The study was done among practitioners from private for-profit clinics and hospitals in Kampala present between March and July 2020 who gave informed consent. Practitioners included medical officers, specialist doctors, clinical officers and laboratory technicians. Laboratory technicians in Uganda are health workers who undergo training in laboratory work and diagnostics, may train for one, two or three years to acquire a certificate, a diploma or degree in laboratory science respectively from a recognized allied health professional institute in case of certificates and diploma and a recognized university in case of degree level qualification. Clinical officers undertake a three years training in an allied health professional's college to acquire a diploma in clinical medicine. They are mainly employed at lower-level health facilities in government aided health facilities but a number of private for health facilities employ them. The doctors in Uganda have at least a bachelor of medicine and bachelor of surgery from a recognized university. In the structure of ministry of health- Uganda, clinical officers and laboratory technicians along with other health workers with certificate or diploma level qualifications, are referred to as allied health professionals.

The study excluded practitioners that were attending to emergency or theatre duties on interview day.

## Sample size estimation

The sample size for assessment of practitioners' diagnosis and treatment practices was estimated using a formula for single proportions with adjustment for clustering [19]. We assumed 95% level of confidence, 5% sampling error, proportion of practitioners having standard diagnosis and treatment practices of 24% [9], and a design effect of 2 to cater for clustering. This gave an estimated sample size of 561. Following adjustment for a non-response of 10%, a higher sample size of 630 was obtained and these were the participants involved in the study.

## Sampling

The number of PFPs to be visited in each division of the city was calculated using the proportion of PFPs in that division to the total PFPs in Kampala as per the health facilities master list provided by ministry of health. The particular facilities to participate were selected randomly using a computer-generated list of random numbers and using the list of private for-profit health facilities from health facilities master list for each division as a sampling frame. One practitioner was selected to participate from each facility. Proportionate sampling was used in selecting practitioners whereby for every one doctor interviewed, 5 allied health professionals (clinical officers and laboratory technicians) were interviewed. For each facility the name of the participant was randomly selected from the names of eligible practitioner's present on the day of the interview.

## Study variables

**Independent variables.** Health facility factors: Type of facility (outpatient only versus outpatient and inpatient), qualification of the facility director, involvement with NTLP,

availability of equipment for diagnosis including gene expert machine or microscope, supervision by NTLP, years of existence of the facility.

Practitioner factors: Age of the practitioner, qualification, whether they are completely in private practice or part-timing between private and public facilities, years in practice, knowledge of diagnosis and treatment of TB, number of years since last TB related training, having a copy of TB guidelines in the facility that they refer to when need arises, income earned by the practitioner, number of presumptive TB cases seen in the last one year confirmed from facility records, common source of information for updating their knowledge on TB, quality of the relationship of NTLP with private for profit facilities from their point of view.

**Dependent variable.** Practices of diagnosis and treatment of tuberculosis in this study was a binary outcome scored as standard or substandard practices using the ISTC guidelines. The Uganda national tuberculosis management guidelines are adopted from the ISTC guidelines [20]. A practitioner's practices were scored standard if they provided 75% correct responses overall in the section of the questionnaire that was assessing practices and anything less was scored as substandard practices. The choice to use a cut-off of 75% was informed by literature where the ISTC guidelines had been previously used and the fact that all practitioners were assessed on both TB diagnostics and treatment regardless of their qualifications [21].

**Data collection tools and procedures.** A structured questionnaire was used to collect the data and it had four parts. The first part was on the socio-demographic information and professional characteristics of practitioners. The second part assessed knowledge of the practitioners regarding pulmonary TB diagnosis and treatment. The third part had questions assessing practices and had been adopted from a previous study that assessed knowledge and practices among private practitioners using ISTC guidelines [22]. The fourth part was concerning facility-related characteristics of the participants. The questionnaires were self-administered but in the presence of the research assistant who waited for the practitioners to complete it.

**Data management.** Quantitative data were entered in Epidata version 4.4.1.0. Data was entered in a computer with restricted access and backed up at all points to mitigate loss. Completeness and correctness of the data was checked at the point of data collection.

**Data analysis.** Analysis was done in STATA version 14 College Station, Texas. Participants' characteristics were described using proportions for categorical variables and continuous variables were summarized using means and standard deviations for normally distributed ones or medians and interquartile ranges (IQR) for skewed variables.

We computed the proportion of practitioners with standard practices. Generalized linear models with log Poisson link was used to analyze the relationship between practitioners' practices and the independent variables. At bivariate analysis factors with a p-value of less than 0.2 were taken to multivariate analysis. To improve the precision of estimates, the data was declared as survey data and the analysis done through survey data analysis window of STATA version 14 College Station, Texas and clustered robust standard errors were used to cater for clustering in the level of qualifications of participants. Variables were considered to have a significant association if their $p < 0.05$. Interaction was assessed first using chunk test and then by manual dropping of interaction terms basing on their significance in the model. Confounding was assessed by comparing prevalence ratios in adjusted and unadjusted models and a variable was considered a confounder if it produced a difference of 10% or above.

**Ethical considerations.** Ethical approval was obtained from Makerere University School of Medicine Research Ethics Committee approval number #REC REF2020-026. Administrative approval was obtained from the office of the Director of public health and environment at Kampala Capital City Authority. Informed consent was sought from the private practitioners that participated in the study. Using serial numbers instead of names ensured confidentiality of the participants.

# Results

## Characteristics of participants

A total of 630 practitioners from PFP facilities within Kampala participated in the study as summarized in Table 1 below. The median age of participants was 29 years (IQR = 26–34), majority were only working in private facilities (78.9%), and the median number of years in practice was 4 years (IQR = 2–7).

The median number of years since the last TB training was 1 year, and the median number of presumptive TB cases seen in the last one years was 3 cases. Overall, 16.8% of the practitioners that participated in the study did not know about the existence of the national tuberculosis and leprosy program and its coordination of the TB clinical services.

**Table 1. Sociodemographic characteristics of practitioners from private for-profit clinics and hospitals in Kampala.**

| Characteristic | Frequency (n) or Median (IQR) | Percentage (%) |
|---|---|---|
| **Age, IQR, years** | 29(26–34) | |
| **Qualification** | | |
| Clinical officer | 245 | 38.9 |
| Laboratory technicians | 268 | 42.5 |
| Doctors | 117 | 18.6 |
| **Level of qualification** | | |
| Certificate | 101 | 16 |
| Diploma | 392 | 62.2 |
| Degree | 137 | 21.8 |
| **Type of employment** | | |
| Only in private facility | 497 | 78.9 |
| Part-timing between private and public facilities | 133 | 21.1 |
| **Number of years in practice, IQR, years** | 4(2–7) | |
| **Income** | | |
| Satisfying | 155 | 24.6 |
| Not satisfying | 475 | 75.4 |
| **Numbers of years since last TB related training** | 1(0–3) | |
| **Knowledge of TB** | | |
| Unacceptable | 347 | 55.1 |
| Acceptable | 283 | 44.9 |
| **Has a copy of TB diagnosis and treatment guidelines** | | |
| No | 352 | 55.9 |
| Yes | 278 | 44.1 |
| **Number of presumptive TB cases seen in the last one year** | 3(0–6) | |
| **Source of TB related information and updates** | | |
| Others | 496 | 78.7 |
| NTLP | 134 | 21.3 |
| **Description of extent of NTLP involvement with PFPs** | | |
| Good enough | 121 | 19.2 |
| Not good enough | 403 | 64 |
| I don't know about it | 106 | 16.8 |

From Table 1 above, the median age of the participants was 29(IQR: 26–34) years, 55.1% had an unacceptable level of knowledge and the number of presumptive TB cases seen by a practitioner in the last one year was 3(IQR:0–6).

Overall, 62.4% of the facilities had both inpatient and outpatient facilities, 60.8% have been in existence for at least 5 years, the majority (85.7%) had no working relationship with NTLP at all and 63.7% had a microscope in-house that can be used for TB diagnosis, Table 2.

## Practices of TB diagnosis and treatment of practitioners

Overall, the proportion of practitioners with standard practices was 46.2% (95% CI: 26.6% to 67.1%).

From Table 3, majority of the private practitioners were defining a presumptive TB case correctly in their practice (94.1%), but only a small proportion (21.1%) correctly identified all the possible sources of specimens for TB diagnosis in children with intrathoracic TB. Although 70.4% were practicing the right number of phases of treatment in first time TB patients, only 32.4% were prescribing the right drugs and knew the right names of the drugs for each phase in a first-time patient without resistance.

## Bivariate analysis for factors associated with practices of diagnosis and treatment of TB among the private practitioners in Kampala

From Table 4 above, variables that had a p-value less than 0.2 were taken to multivariate analysis and these included: age, qualification, level of qualification, type of employment and income, number of years since last TB related training, knowledge of TB, and having a copy of TB guidelines.

From Table 5 above, variables with a p-value of less than 0.2 were taken to multivariate analysis and include director's qualification, works with NTLP, supervision by NTLP and equipment for TB diagnosis.

**Table 2. Facility-related characteristics practitioners from private for-profit clinics and hospitals in Kampala.**

| Characteristic | Frequency (n) | Percentage (%) |
|---|---|---|
| **Type of facility** | | |
| Outpatient only | 237 | 37.6 |
| Both outpatient and inpatient | 393 | 62.4 |
| **Director's qualification** | | |
| Clinical officer | 234 | 37.1 |
| Medical officer | 267 | 42.4 |
| Specialist | 129 | 20.5 |
| **Years in existence** | | |
| <5years | 247 | 39.2 |
| ≥5years | 383 | 60.8 |
| **Works with NTLP** | | |
| No | 540 | 85.7 |
| Yes | 90 | 14.3 |
| **Supervision by NTLP** | | |
| Never | 552 | 87.6 |
| Yearly | 39 | 6.2 |
| Quarterly | 39 | 6.2 |
| **Equipment for TB diagnosis** | | |
| None | 173 | 27.5 |
| Microscope only | 401 | 63.7 |
| Microscope plus x-ray machine or gene-Xpert machine or all three | 56 | 8.9 |

**Table 3. Specific practices of diagnosis and treatment of TB among private practitioners in Kampala.**

| ISTC guidelines | Standard practice n (%) |
|---|---|
| **Standards for diagnosis** | |
| **Standard 2** | |
| Defining a presumed TB case | 593(94.1) |
| **Standard 3** | |
| Sputum collection for TB diagnosis | 435(69.1) |
| Preferred method for TB diagnosis in people living with HIV | 502(79.7) |
| **Standard 6** | |
| All possible sources of specimen for TB diagnosis in children with intrathoracic TB | 133(21.1) |
| **Standards for treatment** | |
| **Standard 8** | |
| Phases of treatment in a first time TB patient | 261(70.4) |
| Drugs for previously untreated patients without risks for drug resistance | 120(32.4) |
| Duration of treatment for each phase | 202(54.5) |
| **Standard 10** | |
| Monitoring response to treatment at the end of the initial phase | 187(50.4) |
| **Standard 13** | |
| Availability of TB registers in the facility | 132(35.6) |
| Names of at least one TB register used in our setting | 63(17.0) |

## Multivariate analysis for factors associated with practices of diagnosis and treatment of TB among the practitioners

From Table 6 above, factors that were significantly associated with practices of TB diagnosis and treatment among practitioners from private for-profit clinics and hospitals in Kampala at multivariate analysis were qualification of the practitioner being a laboratory technician (PR = 2.729, $p< 0.001$) or doctor (PR = 1.204, $p< 0.001$), degree level of qualification (PR = 1.095, $p = 0.021$), quarterly supervision by the national TB program (PR = 1.339, $p = 0.023$), acceptable knowledge of the practitioner about TB (PR = 1.756, $p <0.001$.

Throughout our analysis, we chose to use prevalence ratios instead of odds ratios because our prevalence of standard practices was high at 46.2%, making the prevalence ratio a better estimate of the measure of association than an odds ratio.

## Discussion

Our findings show that 46.2% of practitioners from private for-profit health facilities in Kampala have self-reported standard practices of TB diagnosis and treatment. At least half of the PPs have practices that are not in line with the international standards of TB care set by the WHO from which the Uganda NTLP derives its guidelines. Guidelines-directed TB management and diagnosis provides an efficient way of managing and controlling TB both on a local and global scale. But with such less than average standards, it means patients with TB presenting in private for-profit facilities in Kampala may receive less than standard management. This definitely fuels the TB epidemic in Uganda, and has unpleasant implications for global health.

Overall, PPs in urban Uganda have better diagnostic standards 64.6 (26.6–67.1) % than treatment standards 40.9 (8.4–83.9) % and this varied across cadres. Notable was the significant variation in treatment practices among prescribing cadres with proportions of standard treatment practices of 31.4% and 60.7% among the clinical officers and doctors respectively. Though this may be explained by the difference in cadres, for a disease of global health

**Table 4. Bivariate analysis for sociodemographic characteristics of private practitioners in Kampala.**

| Characteristic | Standard practices | | PR (95% CI) | P-value |
|---|---|---|---|---|
| | Yes n (%) | No n (%) | | |
| **Age** | | | 0.994(0.990–0.998) | 0.004 |
| **Qualification** | | | | |
| Clinical officer | 63(25.7) | 182(74.3) | 1 | |
| Laboratory technician | 179(66.8) | 89(33.2) | 2.597(2.229–3.027) | < 0.001 |
| Doctor | 49(41.9) | 68(58.1) | 1.629(1.576–1.683) | < 0.001 |
| **Level of qualification** | | | | |
| Certificate | 62(61.4) | 39(38.6) | 1 | |
| Diploma | 161(41.1) | 39(58.9) | 0.669(0.669–0.669) | < 0.001 |
| Degree | 68(49.6) | 39(50.4) | 0.809(0.809–0.809) | < 0.001 |
| **Type of employment** | | | | |
| Only in private | 234(47.1) | 263(52.9) | 1 | |
| Part-timing between private and public facilities | 57(42.9) | 76(57.1) | 0.910(0.838–0.989) | 0.026 |
| **Years in practice** | | | 0.993(0.979–1.008) | 0.36 |
| **Income** | | | | |
| Satisfying | 66(42.6) | 89(57.4) | 1 | |
| Not satisfying | 225(47.4) | 250(52.6) | 1.112(0.970–1.276) | 0.128 |
| **Number of years since last TB related training** | | | 0.979(0.964–0.995) | 0.008 |
| **Knowledge of TB** | | | | |
| Unacceptable | 134(38.7) | 212(61.3) | 1 | |
| Acceptable | 157(55.3) | 127(44.7) | 1.427(1.149–1.774) | < 0.001 |
| **Having copy of TB guidelines** | | | | |
| No | 147(41.8) | 205(58.2) | 1 | |
| Yes | 144(51.8) | 134(48.2) | 1.240(0.969–1.587) | 0.087 |
| **Number of presumptive TB cases seen in the last one year** | | | 1.003(0.996–1.011) | 0.405 |
| **Source of TB related information or updates** | | | | |
| Others | 225(45.4) | 271(54.6) | 1 | |
| NTLP | 66(49.3) | 68(50.7) | 1.086(0.808–1.458) | 0.585 |
| **Description of the extent of NTLP involvement with PFPs** | | | | |
| Good enough | 59(48.8) | 62(51.2) | 1 | |
| Not good enough | 181(44.9) | 222(55.1) | 0.921(0.692–1.225) | 0.572 |
| I don't know about it | 51(48.1) | 55(51.9) | 0.987(0.776–1.254) | 0.913 |

significance like tuberculosis, it is important to rally and train the concerned cadres to achieve the highest standard of management and control.

Though these practices may have improved compared to almost two decades back, where only 24% of the PPs were prescribing standard treatment regimens [9], this is still less than standard and exposes the gaps that still need to be bridged in the fight against TB. Not to mention that since these were self-reported practices, these proportions may be an over estimate meaning the gap may even be larger than estimated in this study. Notably, since 2001 when a study compared public and private practitioners in their appropriateness of TB management in Kampala [9], it has been almost two decades of wide spread activities in TB care and more portals for access to information and therefore this proportion is expected to be much higher.

Knowledge gaps were evident, only small proportion (21.1%) correctly identified all the possible sources of specimens for TB diagnosis in children. Not to mention that only 32.4 (27.8–37.3) % of the PPs were prescribing the right drugs and knew the right names of the

**Table 5. Bivariate analysis for facility related characteristics associated with practices of diagnosis and treatment of TB among private practitioners in Kampala.**

| Characteristic | Standard practices | | PR (95%CI) | P-value |
|---|---|---|---|---|
| | yes n (%) | no n (%) | | |
| **Type of facility** | | | | |
| Outpatient only | 107(45.1) | 130(54.90) | 1 | |
| Both outpatient and inpatient facility | 184(46.8) | 209(53.2) | 1.037(0.914–1.177) | 0.572 |
| **Director's qualification** | | | | |
| Clinical officer | 101(44.4) | 130(55.6) | 1 | |
| Medical officer | 129(48.3) | 138(51.7) | 1.087(0.995–1.188) | 0.065 |
| Specialist | 58(45.0) | 71(55.0) | 1.012(0.766–1.385) | 0.935 |
| **Years in existence** | | | | |
| ≤5years | 113(45.8) | 134(54.2) | 1 | |
| >5years | 178(46.5) | 205(53.5) | 1.016(0.889–1.161) | 0.818 |
| **Works with NTLP** | | | | |
| No | 240(44.4) | 300(55.6) | 1 | |
| Yes | 51(56.7) | 39(43.3) | 1.275(1.246–1.305) | < 0.001 |
| **Supervision by NTLP** | | | | |
| Never | 243(44.0) | 309(56.0) | 1 | |
| Yearly | 20(51.3) | 19(48.7) | 1.165(0.748–1.813) | 0.499 |
| Quarterly | 28(71.8) | 11(28.2) | 1.631(1.065–2.497) | 0.024 |
| **Equipment for TB diagnosis** | | | | |
| None | 76(43.9) | 97(56.1) | 1 | |
| Microscope | 181(45.1) | 220(54.9) | 1.027(0.799–1.322) | 0.833 |
| Microscope plus x-ray or Xpert machine or both | 34(60.7) | 22(39.3) | 1.382(0.886–2.156) | 0.154 |

drugs for each phase in a first-time patient without resistance. Later on, PPs with an acceptable level of knowledge about TB significantly had standard practices (PR = 1.756, p-value<0.001) unlike their counterparts with an unacceptable level of knowledge. In this study, 44.9% of the PPs had an acceptable level of knowledge, this was lower than in an earlier study in Kampala

**Table 6. Factors associated with practices of TB diagnosis and treatment among private practitioners in Kampala.**

| Characteristic | PR | 95% CI | P-value |
|---|---|---|---|
| **Qualification** | | | |
| Clinical officer | 1 | | |
| Laboratory technician | 2.729 | 2.452–3.037 | < 0.001 |
| Doctor | 1.145 | 1.106–1.184 | < 0.001 |
| **Level of qualification** | | | |
| Certificate | 1 | | |
| Diploma | 0.900 | 0.866–0.935 | < 0.001 |
| Degree | 1.095 | 1.014–1.184 | 0.021 |
| **Supervision by NTLP** | | | |
| Never | 1 | | |
| Yearly | 0.986 | 0.685–1.421 | 0.941 |
| Quarterly | 1.339 | 1.041–1.721 | 0.023 |
| **Knowledge of TB** | | | |
| Unacceptable | 1 | | |
| Acceptable | 1.756 | 1.649–1.871 | < 0.001 |

where 55% of the PPs that participated had an acceptable level of knowledge [9]. Similar findings have been reported in other studies, in Indonesia, 96% of the participants lacked knowledge about the diagnosis of TB in children and only part of them (49%) was able to correctly indicate when TB therapy should be started [23] and the findings were not different in Lesotho [15]. These knowledge gaps eventually affect the efforts to implement TB control activities and the agenda to eradicate TB locally and globally. National TB programs need to expand and target all practitioners, public or private, and improve their knowledge because in a high TB burden setting like Kampala, a practitioner is bound to receive a TB patient any time.

For successful implementation of TB control strategies, some practices are paramount. One such practice would be maintaining proper registry of patients for TB screening, diagnosis and treatment; these are key to contact tracing and following up patients for treatment completion. Unfortunately, only 35.6% of the PPs had at least one TB register in their facility and only 17% could name at least one name of a TB register that should be found in a facility involved in TB care. This means that there is poor record keeping and follow up of TB patients in PFP yet this is an important aspect of TB management and control. Such findings were also registered in a cross-sectional study in Addis Ababa at even higher rates, 80% of the PPs did not keep a TB register [24]. It is important that PPs are rallied to recognize the significance of record keeping in TB control programs so they can give it the appropriate attention.

With such small proportions of PPs having standard practices, it means that Uganda may not meet goals set by the WHO, 95% reduction in TB deaths and 90% reduction in TB incidence, by 2035 as stated in the end TB strategy [25]. Moreover, this strategy specifies one of the key components to achieving these goals as engagement of all public and private health care providers. But this is not the case as only 14.3% of the PPs in this study were found in facilities that are involved with the NTLP. This non-involvement could go a long way to explain the substandard practices among these PPs as they are not exposed to the periodic trainings, updates in TB care and supervision. Not to mention that they do not receive reagents, TB registers and anti-TB drugs like the public facilities do which may demotivate them and even cause disinterest in making effort to keep up to date with standard practices. The non-involvement of PFPs in NTLP activities in such a high TB burden country therefore remains a major barrier in the fight against TB.

Qualifications of the practitioners were highly associated with standard practices with doctors (PR = 1.145, p-value<0.001) more likely to have standard practices than clinical officers despite both being prescribing clinicians and this was also reflected through the level of qualification as measured by the highest award achieved by the practitioner with degree level PPs more likely to have standard practices (PR = 1.095, p-value = 0.021) than the certificate and diploma level PPs. These findings were similar to those found by Nshuti et al almost two decades ago where degree level practitioners were more likely to provide appropriate TB care (odds ratio 4.8) [9]. In a cross sectional study of PPs in the Philippines, a diagnosis of TB was three times more likely from specialized physicians (chest, internal medicine and infectious diseases) than from general practitioners or family physicians (OR 3.55, $P<0.001$) [26]. As a country dealing with a high burden of TB, maybe it is time that we that we rally highly trained practitioners, involve them in TB control and management activities, equip them with the resources to train and supervise those below them as a way to improve standards of TB management. Quarterly supervision by the national TB program was associated with standard practices (PR = 1.339, p-value = 0.023). These findings have been consistent in other studies in Indonesia [23]. This goes on to emphasize the need and benefits that could arise from wholesome involvement of all practitioners in national TB programs regardless of whether they are public or private. Moreover, studies of PPs elsewhere have shown that PPs whose facilities are involved with the national TB programs are more likely to have under gone training compared

to those whose facilities are not involved [22] and such trainings have been associated with standard practices [14,22]. In neighboring Nairobi, a study of private providers concluded that they can be engaged to provide TB-HIV care conforming to national standards but it meant providing diagnostics, drugs, training and as well as supervising these providers [27]. In India, involving private sector doctors/staff during quarterly/monthly review meetings and providing all technical inputs and support was at the center of improving private sector engagement in national TB programs [28]. All these studies, pointed to a wholesome involvement with programed interaction between national TB programs and private facilities. Such strategies could position the PPs in Uganda, the biggest provider of health care in urban settings, to provide standard TB control and management.

According to WHO, the keys to motivating private providers to conform with the International Standards of TB Care (ISTC) have been well articulated by implementers and include building respectful relationships, understanding providers' need to retain patients and safeguard their reputations, and acknowledging the importance of revenues but also access to new technologies and professional development [8].

## Strengths and limitations of the study

Our study reflected a large number of private practitioners from private for-profit facilities and from across all cadres, which makes its findings relatable and representative of Kampala. However, these were self-reported knowledge and practices of TB diagnosis and treatment which may have been affected by information bias. There may have been random error given the wide confidence intervals which could affect generalizability.

## Conclusions

The practices of practitioners from the private for-profit hospitals and clinics in Uganda's capital city were less than average at only 46.2% overall standard practices. This means that the largest provider of health services in Kampala and generally Uganda, private for-profit facilities, provides substandard services for TB diagnosis and treatment. Moreover, a large number of the facilities do not have a working relationship with the national TB and leprosy program which oversees TB care services in the country. This presents a big implementation challenge for delivery of TB services since majority of the population is not being targeted for TB diagnosis and treatment. Moreover, delivery of TB management and control services has been further disrupted by emerging epidemics like COVID-19. Without a completely unified fight between private for-profit healthcare providers in Uganda and the national TB program, a number of TB cases will not be linked to timely and standard diagnosis and treatment, and TB will remain a threat not only to the health of Ugandans but also to global health for decades to come. As a country, the priorities in the fight against TB must focus on involving as many private for-profit healthcare providers in the activities of the national TB program as possible.

## Acknowledgments

I acknowledge the contribution of Kampala Capital City Authority (KCCA) and the directors of the participating facilities for granting our research team access to the facilities as well as the health workers who participated in the study and the staff of the clinical epidemiology unit at Makerere University College of Health Sciences for their mentorship support.

## Author Contributions

**Conceptualization:** Judith Amutuhaire Ssemasaazi, Joan N. Kalyango, Achilles Katamba.

**Data curation:** Judith Amutuhaire Ssemasaazi.

**Formal analysis:** Judith Amutuhaire Ssemasaazi, Felix Bongomin, Rebecca Akunzirwe, Ronald Muganga Kivumbi, Joan N. Kalyango.

**Funding acquisition:** Judith Amutuhaire Ssemasaazi.

**Investigation:** Judith Amutuhaire Ssemasaazi.

**Methodology:** Judith Amutuhaire Ssemasaazi, Rebecca Akunzirwe, Joan N. Kalyango.

**Project administration:** Judith Amutuhaire Ssemasaazi, Cherop Adolphus.

**Software:** Felix Bongomin.

**Supervision:** Judith Amutuhaire Ssemasaazi, Ezekiel Mupere, Achilles Katamba.

**Visualization:** Joan Rokani Bayowa, Joan N. Kalyango.

**Writing – original draft:** Judith Amutuhaire Ssemasaazi.

**Writing – review & editing:** Felix Bongomin, Rebecca Akunzirwe, Joan Rokani Bayowa, Emmanuel Ssendikwanawa, Cherop Adolphus, Ronald Muganga Kivumbi, Joan N. Kalyango, Phiona Ekyaruhanga, Achilles Katamba.

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
