## [Decision Letter · Decision Letter 0]

13 Feb 2023

PONE-D-22-31914Private Practitioners’ practices and implementation challenges for tuberculosis management and control in a city largely served by the private health sector in UgandaPLOS ONE

Dear Dr. Amutuhaire Ssemasazi,

Thank you for submitting your manuscript to PLOS ONE. After careful consideration, we feel that it has merit but does not fully meet PLOS ONE’s publication criteria as it currently stands. Therefore, we invite you to submit a revised version of the manuscript that addresses the points raised during the review process.

We look forward to receiving your revised manuscript.

Kind regards,

Lara Vojnov

Academic Editor

PLOS ONE

Journal Requirements:

This original research was partly funded by a student research scholarship awarded to Judith Amutuhaire-Ssemasaazi by the Africa Center of Excellence in Materials, Product Development and Nanotechnology (MAPRONANO ACE) Makerere University. Details about this student support fund can be accessed at https://mapronano.mak.ac.ug/

This original research was partly funded by a student research scholarship awarded to Judith Amutuhaire-Ssemasaazi by the Africa Center of Excellence in Materials, Product Development and Nanotechnology (MAPRONANO ACE) Makerere University. Details about this student support fund can be accessed at https://mapronano.mak.ac.ug/

I acknowledge the Africa Center of Excellence in Materials, Product Development and Nanotechnology (MAPRONANO ACE) Makerere University for providing funding for this research and the staff of the clinical epidemiology unit at Makerere University College of Health Sciences for their mentorship support.

However, funding information should not appear in the Acknowledgments section or other areas of your manuscript. We will only publish funding information present in the Funding Statement section of the online submission form. 

This original research was partly funded by a student research scholarship awarded to Judith Amutuhaire-Ssemasaazi by the Africa Center of Excellence in Materials, Product Development and Nanotechnology (MAPRONANO ACE) Makerere University. Details about this student support fund can be accessed at https://mapronano.mak.ac.ug/

Additional Editor Comments:

Thank you for submitting your manuscript to PLoS One. After careful consideration, we'd like you to very carefully review and reflect the Reviewer Comments, particularly those of Reviewer #1, prior to returning the manuscript for further consideration. Only if all suggested changes can be met will we consider the manuscript for publication.

Reviewers' comments:

Reviewer's Responses to Questions

**Comments to the Author**

1. Is the manuscript technically sound, and do the data support the conclusions?

Reviewer #1: No

Reviewer #2: Yes

2. Has the statistical analysis been performed appropriately and rigorously? 

Reviewer #1: No

Reviewer #2: Yes

3. Have the authors made all data underlying the findings in their manuscript fully available?

Reviewer #1: Yes

Reviewer #2: Yes

4. Is the manuscript presented in an intelligible fashion and written in standard English?

Reviewer #1: No

Reviewer #2: Yes

5. Review Comments to the Author

Reviewer #1: The manuscript presents a study of a large number of private providers in Kampala and their involvement in the TB program which is a needed area of research since most of the work in the PPM space has focused on Asian settings. Uganda has one of the larger private sectors among African countries, and the results of PPM studies could certainly be useful for the TB community.

I felt reading the manuscript that the authors had not done as much review and thought as should have gone into a journal submission however and believe that many things should be addressed before it could be considered for publication.

There are numerous typos, spacing issues, and data inconsistencies (ie number of TB deaths is different in abstract and intro, rounding those deaths to three digits, ) and in the text throughout the document which make reading the manuscript difficult. The authors use stigmatizing language like TB suspect that should not be used in any publication in these days.

I found that in general the manuscript read like a very detailed academic thesis that had been turned into a manuscript, which should be congratulated, but only after careful editing.

The discussion alone is 16 pages long which is far too much text for any publication, and goes through in great detail about too many of the individual findings.

Here are a few areas for consideration as the authors review the manuscript.

Introduction:

In general what is missing in the introduction is more of a sense of the PPM work that has been done in Uganda. What models have been used by the NTP – what has worked in the past – there have been Global Fund supported approaches for PPM. What is the private sector allowed to do? Can they prescribe TB drugs? Are they available for purchase with or without a prescription? What proportion of TB in Uganda comes from PPM? How much is the current engagement level? This information is very important to understand the context and the results.

Line 60 – missing cases is not defined, and but then 2.9 million should be the difference between 10.6 and the notifications – which in 2021 was 6.4 although not stated. This difference is then 4.2m. Line 61 says 60-70% of first line health services – but not necessarily TB, as we find out later - in reference 8 the study is done in Uganda showing that 895 of 1351 people actually sought care in the public sector for TB. The references are not well structured in general.

Line 68 DOTS is directly observed therapy shortcourse – but DOTS the old WHO branded TB program was not directly observed therapy program – it was a brand for TB control – they were confusing, but different – and now the DOTS label has been totally dropped. Line 72 - What does ‘involvement in TB programs’ mean separate from the other parts of the sentence? A PPM approach in general?

Line 75 – the number of private facilities has increased is quite vague.

Methods Line 89 the data is from 2012 but earlier there have been large increases? Can the authors provide more details? Isnt there another source of data? Later the health facility lists are mentioned which would be more useful to say up front. Also, providing only proportions here is not as helpful as some figures, 100 labs, 200 clinics, 30 hospitals etc. There are major differences between types of facilities as well so more detail would help.

Line 91 Highest proportion compared to what? Other districts? By population?

It is unclear why there is a selection of very different types of workforce (lab tech, clinical officers and doctors) it is also not clear why the 5:1 ratio for selecting was used (line 122). It might be a good idea to define the difference between a doctor and clinical officer for those unfamiliar with those cadres. I also found the details of the sampling framework too much for this type of study. Why even ask the lab techs about treatment data. They probably should not be expected to know about it. Line 105 and 113 talk about two different sample sizes. And the Kish Leslie formula – is not explained? 117-118 again – without the total numbers of facilities the reader is a bit in the dark. The sampling framework on paper is a strong exercise which was done quite well for a nice random sample of the practitioners, but for the outcome of the stud, it seems like maybe not the most important way to do things since generally decisions about diagnosis and treatment may be made through consultation (and also lab techs wont treat) can clinical officers prescribe? By just capturing an individual’s knowledge about a relatively rare disease, what are we able to say about the private sector overall?

Line 131 – what relevance is the degree of the director to the study?

132 – should be GeneXpert – and I am not sure this is particularly relevant either? How many have them? It would only be in very high volume facilities I imagine?

136 – why is knowledge of TB an independent variable? Having a copy of the Guidelines would mean there is already some engagement, and likely training – are those correlated?

The sections on independent and dependent variables are very long and detailed for a manuscript – I would refer to an online annex to help the reader understand and reduce text and just describe briefly.

154 – how can a semi structured questionnaire be self administered?

Line 156 talks about key informants which is the first time they are mentioned – but it is not clear who they are.

What analysis program was used for the interviews? All done by hand? Seems strange.

This is also part of the submission that confused me. In my opinion, the two strategies for data collection are confusing, and not complimentary. They make the submission incredibly long and difficult to follow and I am not sure they add so much. I would suggest separating them into two papers.

Again the data analysis section, there is far to much detail for a peer reviewed publication, and it seems more of a thesis document.

Results: Same as above – too much detail – too many tables with similar types of data. Variables that are not defined etc. What is a certificate mean in table 1? What does Satisfying mean? In table 2 does the number of presumptive come from their memory? Or is it abstracted from a register and verified? This is a major difference. How was the last NTP training years measured? Was it corroborated with NTP data? The question on description of NTP involvement seems incredibly bias and not very useful – in a questionnaire about TB? If they see a median of 3 people a year, is it worth having the NTP spend time and energy to train and engage them? Table 3 – the denominator for supervision is all interviewed, but this really should only be those who are engaged – otherwise, why would the NTP supervise? The data on microscopes is interesting with so many having them – are they used for TB or other diseases? That would be a more useful question. Also GeneXpert not gene-xpert

Lines 213 and 215 use of verb tense, ‘About 62.4% is very specific – consider Overall,….

The use of the ISTC GL for measurement – are they still valid? Is this was Uganda uses? Maybe make a mention of this in the methods? I found the question on children difficult. I would guess that most clinicians in public facilities wouldn’t not know that either. How many questions make up the standards? 10? Why chose 75% as the cutoff? These are unclear. In the table 4 Standards for Treatment is not bolded like diagnosis is. For standard 13 – again – why use all for the denominator when so many don’t use these registers? It isn’t a fair comparison. Knowing the drug names as a lab tech is not very important, but for a prescriber it is. If a private clinic cannot prescribe in Uganda unless they are NTP certified, then it is less important – and hence the suggestion to add these kinds of details in the introduction.

Having 9 tables in this results section is far to much. And then we read through all of the individual thoughts on different aspects of the TB situation, I suggest splitting these into two papers.

The discussion, as stated above, was simply far too long. Each result of the standards cannot be discussed and there was too little thinking about what it actually means for Uganda which is far more important than comparing results to India or other countries. Line 638 – why reference an out of date TB strategy? Use the End TB Strategy.

I hope the authors take some time to rework the manuscript and I encourage them to publish this work – it is quite comprehensive and can help PPM efforts in Uganda.

Reviewer #2: Dear Author,

The study is of public health importance for achieving the END TB targets. The authors have considerably made good efforts in writing the manuscript.

The following are the minor suggestions:

1. Title: There is a scope to improvise - keep it simple and short

2. The following variables needs operational definition as they are quite subjective - Income, knowledge of TB, extent of NTLP involvement

3. In the methods or data analysis the authors have not explained, why they chose to measure Prevalence ratio.

4. The discussion is more elaborate and general. The emphasis should be on the usefulness of the findings in relevance to the NTLP.

5. Conclusion - There is lot more scope to reduce the sentences and keep it more focused.

Regards,

6. PLOS authors have the option to publish the peer review history of their article (what does this mean?). If published, this will include your full peer review and any attached files.

Reviewer #1: No

Reviewer #2: **Yes: **Sharath Burugina Nagaraja

---

## [Author Response · Author response to Decision Letter 0]

3 Jul 2023

REBUTTAL LETTER RESPONDING TO COMMENTS RAISED BY THE PLOS ONE EDITORIAL TEAM.

10th May 2023

To

Lara Vojnov

Academic Editor

PLOS ONE

Dear Editor,

Reference: PONE-D-22-31914

Thank you for inviting us to submit a revised version of our research manuscript entitled,

‘Private Practitioners’ practices for tuberculosis management in a city largely served by the private health sector in Uganda’. 

We apologize for the delay in resubmission and reaffirm our commitment to contribute to science and global health. We will ensure to submit our next set of revisions in a timely manner.

Kindly receive our rebuttal letter responding to each of the comments raised by the reviewers. We have submitted the revised manuscript with track changes as well as a clean copy.

RESPONSE TO GENERAL COMMENTS FROM THE ACADEMIC EDITOR

General comment #1: Thank you for submitting your manuscript to PLOS ONE. After careful consideration, we feel that it has merit but does not fully meet PLOS ONE’s publication criteria as it currently stands. Therefore, we invite you to submit a revised version of the manuscript that addresses the points raised during the review process.

Response: Thank you for your consideration, we are glad that you think our manuscript has merit and we have revised it to hopefully meet the PLOS ONE publishing standards.

General comment #2. Thank you for stating in your Funding Statement:

This original research was partly funded by a student research scholarship awarded to Judith Amutuhaire-Ssemasaazi by the Africa Center of Excellence in Materials, Product Development and Nanotechnology (MAPRONANO ACE) Makerere University. Details about this student support fund can be accessed at https://mapronano.mak.ac.ug/

Response: We have adjusted our funding statement and included it in the cover letter as guided.

General comment #3. Thank you for stating the following financial disclosure: 

This original research was partly funded by a student research scholarship awarded to Judith Amutuhaire-Ssemasaazi by the Africa Center of Excellence in Materials, Product Development and Nanotechnology (MAPRONANO ACE) Makerere University. Details about this student support fund can be accessed at https://mapronano.mak.ac.ug/. Please state what role the funders took in the study. If the funders had no role, please state: "The funders had no role in study design, data collection and analysis, decision to publish, or preparation of the manuscript." 

Response: The funders had no role in the study and we have included this information in our funding statement and included it in the cover letter.

General comment #4. Thank you for stating the following in the Acknowledgments Section of your manuscript: 

I acknowledge the Africa Center of Excellence in Materials, Product Development, and Nanotechnology (MAPRONANO ACE) at Makerere University for providing funding for this research and the staff of the clinical epidemiology unit at Makerere University College of Health Sciences for their mentorship support.

However, funding information should not appear in the Acknowledgments section or other areas of your manuscript. We will only publish funding information present in the Funding Statement section of the online submission form. 

This original research was partly funded by a student research scholarship awarded to Judith Amutuhaire-Ssemasaazi by the Africa Center of Excellence in Materials, Product Development and Nanotechnology (MAPRONANO ACE) Makerere University. Details about this student support fund can be accessed at https://mapronano.mak.ac.ug/

Response: We have removed all funding statements from the acknowledgment section and the entire manuscript and adjusted in our cover letter.

General comment #5. We note that you have indicated that data from this study are available upon request. PLOS only allows data to be available upon request if there are legal or ethical restrictions on sharing data publicly. For more information on unacceptable data access restrictions, please see http://journals.plos.org/plosone/s/data-availability#loc-unacceptable-data-access-restrictions. 

b) If there are no restrictions, please upload the minimal anonymized data set necessary to replicate your study findings as either Supporting Information files or to a stable, public repository and provide us with the relevant URLs, DOIs, or accession numbers. For a list of acceptable repositories, please see http://journals.plos.org/plosone/s/data-availability#loc-recommended-repositories. We will update your Data Availability statement on your behalf to reflect the information you provide.

Response: There are no restrictions to the data access and we shall avail it in a repository or create a URL.

General comment #6. We note that you have stated that you will provide repository information for your data at acceptance. Should your manuscript be accepted for publication, we will hold it until you provide the relevant accession numbers or DOIs necessary to access your data. If you wish to make changes to your Data Availability statement, please describe these changes in your cover letter and we will update your Data Availability statement to reflect the information you provide.

Response: This is noted and we will make our data set available.

RESPONSES TO REVIEWERS' COMMENTS:

RESPONSES TO COMMENTS FROM REVIEWER #1

Dear Reviewer

Thank you for taking the time to review our manuscript, we have done our best to make sure the manuscript is presented intelligibly, corrected the typing errors and the grammar and we will also provide a minimal data set. Below we have taken time to respond to each of the comments raised. 

Comment #1: The manuscript presents a study of a large number of private providers in Kampala and their involvement in the TB program which is a needed area of research since most of the work in the PPM space has focused on Asian settings. Uganda has one of the larger private sectors among African countries, and the results of PPM studies could certainly be useful for the TB community. I felt reading the manuscript that the authors had not done as much review and thought as should have gone into a journal submission however and believe that many things should be addressed before it could be considered for publication.

There are numerous typos, spacing issues, and data inconsistencies (ie number of TB deaths is different in the abstract and intro, rounding those deaths to three digits,) and in the text throughout the document which make reading the manuscript difficult. The authors use stigmatizing language like TB suspect that should not be used in any publication in these days. 

Response: We have checked through and corrected the data inconsistencies and rounded the TB death statistics to a single decimal point. We have worked on the grammar and spacing to the best of our ability. We have removed the stigmatizing language in favor of more acceptable language.

Comment #2: I found that in general the manuscript read like a very detailed academic thesis that had been turned into a manuscript, which should be congratulated, but only after careful editing.

The discussion alone is 16 pages long which is far too much text for any publication, and goes through in great detail about too many of the individual findings.

Here are a few areas for consideration as the authors review the manuscript.

Introduction:

In general, what is missing in the introduction is more of a sense of the PPM work that has been done in Uganda. What models have been used by the NTP – what has worked in the past – there have been Global Fund supported approaches for PPM. What is the private sector allowed to do? Can they prescribe TB drugs? Are they available for purchase with or without a prescription? What proportion of TB in Uganda comes from PPM? How much is the current engagement level? This information is very important to understand the context and the results.

Response: In Uganda, there has been some level of public-private partnerships in the fight against TB. Most of these have come in the form of non-governmental organizational support to government programs, foreign agencies, grants from global funds, and other funders. Some of these projects include

• The Uganda STOP TB partnership funded by global fund is a collaboration between the National TB and Leprosy Program (NTLP) and non-public TB service providers including civil society organizations, community-based organizations, and private sector/businesses. It was formed in 2004 to support and coordinate nonpublic partners engaged in TB control in Uganda (https://ustp.org.ug/). 

• USAID Defeat TB (Defeat TB) project was funded by USAID and implemented through local partnerships with the infectious disease institute and The AIDS Support Organization (TASO)(https://www.urc-chs.com/projects/defeat-tb/). This five-year project between 2017 -2022, whose main objective was to increase TB case detection and treatment success rates. It collaborated with the ministry of health and the NTP to strengthen existing healthcare systems for TB service delivery in the three districts of Wakiso, Mukono, and Kampala. To maximize their outputs, Defeat TB aimed to identify high volume private facilities and private facilities in TB hot spots and train their health workers in TB case detection and treatment, support integration of TB services into HIV care services, support transportation systems for samples from these facilities and supply the necessary consumables. By the end of the five-year period, Defeat TB had partnerships with 114 private facilities in the three districts and in 2019, 33 active TB cases were detected by these private facilities (https://www.urc-chs.com/news/partnering-with-the-private-sector-to-improve-tb-detection-and-care/). 

• Accelerated Control of TB in Karamoja program (PACT Karamoja) funded by the U.S government and supported by the United States Agency for International Development (USAID) was started in 2020 and runs till 2025. The program aims to address the Karamoja Region’s high TB prevalence and low treatment success rates by supporting locally generated solutions to mobilize health facilities, village health teams, and community members for accelerated screening, testing, identification, and successful treatment and prevention of TB (https://idi.mak.ac.ug/usaid-pact-karamoja/).

• Center for disease control (CDC) using funds from the U.S. government supports the National TB and Leprosy Program (NTLP) through the ministry of health. It has worked with the NTLP since 1991 to increase TB detection and treatment and build capacity for TB treatment t district facilities. It has also worked through non-profit organizations like Baylor Uganda to support TB/HIV treatment services (https://www.cdc.gov/globalhealth/countries/uganda/default.htm).

• The national TB and leprosy program has collaborated with private not-for-profit health facilities in the country, non-profit organizations like the Uganda catholic bureau, and the Uganda protestant bureau. NTLP also collaborates with research and academic institutions like the Infectious Disease Institute (IDI). 

Comment #3: Line 60 – missing cases is not defined, but then 2.9 million should be the difference between 10.6 and the notifications – which in 2021 was 6.4 although not stated. This difference is then 4.2m. Line 61 says 60-70% of first line health services – but not necessarily TB, as we find out later - in reference 8 the study is done in Uganda showing that 895 of 1351 people actually sought care in the public sector for TB. The references are not well structured in general.

Response: We have included data on missing cases from 2021. There were 4.2 million missing cases in 2021(1). We have also included information on the proportion of patients likely to approach private for-profit facilities as the first line when seeking care for TB diagnosis and treatment. A previous study in Uganda show that 21.4% of the participants with a chronic cough first sought care in a private facility and these were also more likely to have a positive sputum gene Xpert (2). We have reorganized the references as well.

Comment #4: Line 68 DOTS is directly observed therapy short-course – but DOTS the old WHO branded TB program was not directly observed therapy program – it was a brand for TB control – they were confusing, but different – and now the DOTS label has been totally dropped. Response: DOTS in line meant Directly Observed Therapy Short course. It has been corrected appropriately in the manuscript.

Comment #5: Line 72 - What does ‘involvement in TB programs’ mean separate from the other parts of the sentence? A PPM approach in general?

Response: Involvement in TB programs in this line was used to refer to the participation of private for-profit facilities in the activities of national TB programs like training, TB diagnosis and treatment, using laboratory referral networks for transporting sputum samples for diagnosis. This was in reference to studies done elsewhere, in these studies, private for-profit facilities that were actively practicing in the activities of the national TB programs had better standards of TB diagnosis and treatment than those that weren’t. In India, attending TB training programs organized by the national TB program was associated with adequate knowledge of TB diagnosis and treatment(3). In Uganda, a public-private partnership between Becton Dickinson and the US President’s Emergency Plan for AIDS Relief, the Uganda National TB Reference Laboratory (NTRL) and National TB and Leprosy Program (NTLP) redesigned the tuberculosis specimen transport network and trained healthcare workers with the goal of improving multidrug-resistant tuberculosis detection. With this intervention, specimen referrals for presumptive multidrug-resistant TB increased by more than 10-fold between 2008 and 2011 improving access to drug susceptibility testing in Uganda. This demonstrated the gains that come with public-private partnerships in the diagnosis of Tuberculosis(4).

Comment #6: Line 75 – the number of private facilities has increased is quite vague.

Response: We modified this to talk about the precise number of private for-profit health facilities. Of the 6397 health facilities in the country, 2795 facilities are private for profit(5)

Comment #7: Methods Line 89 the data is from 2012 but earlier there have been large increases? Can the authors provide more details? Isn’t there another source of data? Later the health facility lists are mentioned which would be more useful to say up front. Also, providing only proportions here is not as helpful as some figures, 100 labs, 200 clinics, 30 hospitals etc. There are major differences between types of facilities as well so more detail would help.

Response: The private health system in Uganda comprises of the Private Not for Profit Organizations (PNFPs), Private Health Practitioners (PHPs) or private for profit and the Traditional and Complementary Medicine Practitioners (TCMPs). Uganda has about 6,937 health facilities and of these 45.16% (3,133) are government owned, 14.44% (1,002) are Private and Not For Profit (PNFP) while the remaining 40.29% (2,795) are Private For Profit (PFP) and 0.10% (7) community-owned facilities(5). Kampala district in particular has approximately 1442 health facilities of which 1367(94%) facilities are private for profit(6). Seventy-six percent (76%) of these PFPs are at the level of health Centre II and health Centre III offering outpatient services to address common diseases like malaria while twelve percent (12%) are at the level of health Centre IV facilities, operating as ‘mini hospitals’ with inpatient care and surgical capabilities(7). A wide variety of services are offered through Uganda’s private health clinics ranging from obstetrics and gynecology, family planning, TB treatment, child health immunizations, sexually transmitted infection (STI) treatments, and safe male circumcisions(7).

Comment #8: Line 91 Highest proportion compared to what? Other districts? By population?

Response: Kampala district contributed the highest incident cases proportion compared to other districts in the country.

Comment #9: It is unclear why there is a selection of very different types of work-force (lab tech, clinical officers and doctors) it is also not clear why the 5:1 ratio for selecting was used (line 122). It might be a good idea to define the difference between a doctor and clinical officer for those unfamiliar with those cadres. 

Response: We interviewed the different types of health work force because these are the main cadres of health workers that play a big role in diagnosis and treatment in Uganda. Private health facilities are manned by mainly these cadres and some private facilities, any one of these cadres may be the only health worker available. The laboratory technicians and clinical officers receive a lower pay than the doctors, as such a number of lower-level private health facilities employ them to reduce their costs. 

Laboratory technicians in Uganda are health workers who undergo training in laboratory work and diagnostics, may train for one, two or three years to acquire a certificate, a diploma or degree in laboratory science respectively.

Clinical officers undertake a three years training in an allied health professional’s college to acquire a diploma in clinical medicine

The doctors in Uganda have their lowest qualification as a bachelor of medicine and bachelor of surgery from a recognized university.

In the structure of ministry of health- Uganda, clinical officers and laboratory technicians along with other health workers with certificate or diploma level qualifications, are referred to as allied health professionals. 

This information has been added in the revised manuscript.

The ratio of 5:1(5 allied health professionals: 1 doctor) for sampling of the participants was used based on the records for the Uganda Medical and Dental practitioners Council which at the number of registered doctors and allied health professionals related in a ratio of 1 doctor for every 5 allied health professionals.

Comment #10: I also found the details of the sampling framework too much for this type of study. Why even ask the lab techs about treatment data. They probably should not be expected to know about it. Line 105 and 113 talk about two different sample sizes. And the Kish Leslie formula – is not explained? 117-118 again – without the total numbers of facilities the reader is a bit in the dark. The sampling framework on paper is a strong exercise which was done quite well for a nice random sample of the practitioners, but for the outcome of the stud, it seems like maybe not the most important way to do things since generally decisions about diagnosis and treatment may be made through consultation (and also lab techs wont treat) can clinical officers prescribe? By just capturing an individual’s knowledge about a relatively rare disease, what are we able to say about the private sector overall?

Response: The sampling became too detailed because we thought it would ensure representativeness. The private health facilities in Kampala city are scattered over the 5 political divisions of the city and thus the first level of sampling ensured that we get a representative sample from each division. In the health facilities, the proportionate sampling was used to ensure that all cadres are represented.

Though laboratory technicians are not primarily trained to prescribe, in Uganda’s private health facilities, it’s very common to find facilities that are owned and run by laboratory technicians who take on the role of prescribing. This is one problem facing Uganda’s private health sector. And of note is that a number of these private facilities do not fulfil all the licensing requirements. For this reason, we included them in the treatment practices assessment.

Clinical officers in Uganda are primarily trained to diagnose and prescribe.

Comment #11: Line 131 – what relevance is the degree of the director to the study?

Response: This part of the variables was formulated with the thought that private for-profit facilities directed by highly qualified health practitioners may have an increased sense of awareness or may tend to interest themselves with diseases of public health concern like tuberculosis.

Comment #12: 132 – should be GeneXpert – and I am not sure this is particularly relevant either? How many have them? It would only be in very high-volume facilities I imagine?

Response: Indeed, GeneXpert may not be in all facilities, more likely to be in high volume settings. In this study we included it not to say that it should be in every facility but to show how many there may be and its association with standard practices of TB management.

Comment #12: 136 – why is knowledge of TB an independent variable? Having a copy of the Guidelines would mean there is already some engagement, and likely training – are those correlated?

Response: Knowledge of TB was treated as an independent variable so as to be to assess its association with standard TB practices. Guidelines in our setting may not necessary be provided by the national TB program. they may be obtained from several sources including pharmaceutical companies, or colleagues in government aided facilities.

Comment #13: The sections on independent and dependent variables are very long and detailed for a manuscript – I would refer to an online annex to help the reader understand and reduce text and just describe briefly.

Response: We have worked to reduce them and we will send some to an online annex

Comment #14: 154 – how can a semi structured questionnaire be self-administered?

Response: It was a structured questionnaire with closed questions and predetermined answers to choose from.

This has been corrected throughout the text.

Comment #15: Line 156 talks about key informants which is the first time they are mentioned – but it is not clear who they are.

What analysis program was used for the interviews? All done by hand? Seems strange. This is also part of the submission that confused me. In my opinion, the two strategies for data collection are confusing, and not complimentary. They make the submission incredibly long and difficult to follow and I am not sure they add so much. I would suggest separating them into two papers.

Response: The key informants were private health practitioners who had been identified during the quantitative interviews as either facility owners or facility in charges that were found enthusiastic about TB programs. The key informant interviews will be re-analyzed and published differently. We have separated the data; we will publish the qualitative in a different paper.

Comment #16: Again, the data analysis section, there is far to much detail for a peer reviewed publication, and it seems more of a thesis document.

Response: We have removed the qualitative section and worked on reducing the content of the tables to condense the size of the manuscript

Comment #17: Results: Same as above – too much detail – too many tables with similar types of data. Variables that are not defined etc. What is a certificate mean in table 1? What does Satisfying mean? In table 2 does the number of presumptive come from their memory? Or is it abstracted from a register and verified? This is a major difference. How was the last NTP training years measured? Was it corroborated with NTP data? The question on description of NTP involvement seems incredibly bias and not very useful – in a questionnaire about TB? If they see a median of 3 people a year, is it worth having the NTP spend time and energy to train and engage them? Table 3 – the denominator for supervision is all interviewed, but this really should only be those who are engaged – otherwise, why would the NTP supervise? The data on microscopes is interesting with so many having them – are they used for TB or other diseases? That would be a more useful question. Also, GeneXpert not gene-Xpert

Response: We have revised our manuscript and reduced the tables.

Here is a description for some of the variables as used in the study.

Certificate level of qualification is a 1-year health professionals training that is offered in Uganda allied health institutions for laboratory technologists and nurses and they are recognized cadre in the health human resource infrastructure.

The income earned by health practitioners in this study was qualitatively categorized as satisfying or not satisfying based on the practitioner’s judgement. We did not attach any quantitative measurements to the variable.

The number of presumptive TB cases seen were abstracted from the facilities record books.

Lat years NTP training was self-reported by the practitioners, it was not corroborated with NTP data.

On the question of microscopes, indeed its quite interesting that a number of practitioners had these in their facilities, which provides an opportunity for diagnosing TB. Though we never followed it with a question to explore whether they utilized them for TB diagnosis, we thought availability of microscopes would provide an opportunity for engagement in TB diagnosis if these facilities receive the right support.

We have corrected the word GeneXpert all through the manuscript

Comment #18: Lines 213 and 215 use of verb tense, ‘About 62.4% is very specific – consider Overall,….

Response: Thank you. The verb has been corrected

Comment #19: The use of the ISTC GL for measurement – are they still valid? Is this was Uganda uses? Maybe make a mention of this in the methods? I found the question on children difficult. I would guess that most clinicians in public facilities wouldn’t not know that either. How many questions make up the standards? 10? Why chose 75% as the cutoff? These are unclear. In the table 4 Standards for Treatment is not bolded like diagnosis is. For standard 13 – again – why use all for the denominator when so many don’t use these registers? It isn’t a fair comparison. Knowing the drug names as a lab tech is not very important, but for a prescriber it is. If a private clinic cannot prescribe in Uganda unless they are NTP certified, then it is less important – and hence the suggestion to add these kinds of details in the introduction.

Response: The Uganda National tuberculosis guidelines were adopted from the ISTC guidelines hence the decision to use these guidelines in our study and we have made a mention of this in the methods section under description of the outcome/dependent variable. The ISTC are 17 standards of which 6 are on diagnosis and the rest are on treatment and control(8). We focused on selected standards that is standard 2,3,6,8,10 and 13 which are fully adopted by the national tuberculosis program as per the Uganda national leprosy and tuberculosis program manual on tuberculosis diagnosis and management. The use of 75% as cut off was based first on a previous study that had used these standards and the fact that both prescribing and non-prescribing health practitioners were exposed to all questions on diagnosis and treatment. 

Comment #20: Having 9 tables in this results section is far too much. And then we read through all of the individual thoughts on different aspects of the TB situation, I suggest splitting these into two papers.

Response: Thank you for your comments, we have removed the qualitative aspect to publish it differently and this has reduced the number of tables and compressed the manuscript.

Comment #21: The discussion, as stated above, was simply far too long. Each result of the standards cannot be discussed and there was too little thinking about what it actually means for Uganda which is far more important than comparing results to India or other countries. Line 638 – why reference an out-of-date TB strategy? Use the End TB Strategy.

I hope the authors take some time to rework the manuscript and I encourage them to publish this work – it is quite comprehensive and can help PPM efforts in Uganda.

Response: We have made effort to re-write our discussion, revised the results and discussion section, reduced the tables and refocused the discussion. Hope it reads better and is more focused and meaningful. 

Thank you for encouraging us to publish this work and we are determined to put the necessary effort to achieve this.

RESPONSES TO COMMENTS FROM REVIEWER #2: 

Dear Reviewer,

Thank you for taking time to review our manuscript and for encouraging us to make an effort to have it published. Below are our responses to each of your comments and we hope they are elaborate enough to qualify our manuscript for publication with PLOS ONE.

General comment: The study is of public health importance for achieving the END TB targets. The authors have considerably made good efforts in writing the manuscript.

Response: This is really encouraging; we appreciate your view of our research and thank you for acknowledging our efforts.

The following are the minor suggestions:

Comment #1: Title: There is a scope to improvise - keep it simple and short 

Response: Thank you for this recommendation. We have revised the title and hope it reads shorter while maintaining the meaning.

Comment #2: The following variables needs operational definition as they are quite subjective - Income, knowledge of TB, extent of NTLP involvement

Response

The variable extent of NTLP involvement with PFPs was a qualitatively measured variable in which practitioners were asked to describe their perception of how much national leprosy and TB program engages with private for-profit facilities. 

Income was also qualitatively assessed; practitioners were asked to tick whether they are satisfied or not satisfied with the income earned from their work at the private health facilities.

Knowledge of TB was assessed using a structured questionnaire and was graded as acceptable if the participant scored 100% or unacceptable if the participant scored less than 100%.

Comment #3: In the methods or data analysis the authors have not explained, why they chose to measure Prevalence ratio.

Response: The choice to use prevalence ratio was based on the fact that our prevalence of standard practices was high, 46.2%. From literature use of prevalence ratio is recommended as a better estimate of the strength of association when the prevalence of the outcome is high, precisely higher than 10% (9, 10).

Comment #4: The discussion is more elaborate and general. The emphasis should be on the usefulness of the findings in relevance to the NTLP.

Response: We have revised our discussion. We hope in its current state, it is more focused and relevant.

Comment #5: Conclusion - There is lot more scope to reduce the sentences and keep it more focused.

Response: Thank you, we have made every effort to keep our manuscript precise and focused, we hope it reads well.

Kind regards,

Judith Amutuhaire-Ssemasaazi (Corresponding author)

References

1. W.H.O. Engaging private health care providers in TB care and prevention: a landscape analysis. 2021. Report No.: 9240027033.

2. Muttamba W, Ssengooba W, Kirenga B, Sekibira R, Walusimbi S, Katamba A, et al. Health seeking behavior among individuals presenting with chronic cough at referral hospitals in Uganda; Missed opportunity for early tuberculosis diagnosis. PloS one. 2019;14(6):e0217900.

3. Basu M, Sinha D, Das P, Roy B, Biswas S, Chattopadhyay S. Knowledge and practice regarding pulmonary tuberculosis among private practitioners. Indian J community Heal. 2013;25(04):403-12.

4. Joloba M, Mwangi C, Alexander H, Nadunga D, Bwanga F, Modi N, et al. Strengthening the Tuberculosis Specimen Referral Network in Uganda: The Role of Public-Private Partnerships. J Infect Dis. 2016;213 Suppl 2(Suppl 2):S41-6.

5. M.O.H. Uganda National Health Facility master list 2018: A complete list of all health facilities in Uganda. 2018.

6. USAID. Health facilty cencus in the Kampala Capital City Authority(KCCA) divisions.; 2017.

7. USAID. Uganda’s private health sector: opportunities for growth.; 2015.

8. Hopewell PC, Pai M, Maher D, Uplekar M, Raviglione MC. International Standards for Tuberculosis Care. The Lancet Infectious Diseases. 2006;6(11):710-25.

9. Martinez BAF, Leotti VB, Silva GSE, Nunes LN, Machado G, Corbellini LG. Odds Ratio or Prevalence Ratio? An Overview of Reported Statistical Methods and Appropriateness of Interpretations in Cross-sectional Studies with Dichotomous Outcomes in Veterinary Medicine. Front Vet Sci. 2017;4:193.

10. Thompson ML, Myers JE, Kriebel D. Prevalence odds ratio or prevalence ratio in the analysis of cross sectional data: what is to be done? Occup Environ Med. 1998;55(4):272-7.

---

## [Decision Letter · Decision Letter 1]

13 Oct 2023

PONE-D-22-31914R1Private Practitioners’ practices for tuberculosis management in a city largely served by the private health sector in Uganda.PLOS ONE

Dear Dr. Amutuhaire Ssemasazi,

Thank you for submitting your manuscript to PLOS ONE. After careful consideration, we feel that it has merit but does not fully meet PLOS ONE’s publication criteria as it currently stands. Therefore, we invite you to submit a revised version of the manuscript that addresses the points raised during the review process.

We look forward to receiving your revised manuscript.

Kind regards,

Lara Vojnov

Academic Editor

PLOS ONE

Journal Requirements:

Additional Editor Comments:

Thank you for the careful review and significantly improved manuscript. Please note some of the remaining comments prior to acceptance.

Reviewers' comments:

Reviewer's Responses to Questions

**Comments to the Author**

1. If the authors have adequately addressed your comments raised in a previous round of review and you feel that this manuscript is now acceptable for publication, you may indicate that here to bypass the “Comments to the Author” section, enter your conflict of interest statement in the “Confidential to Editor” section, and submit your "Accept" recommendation.

Reviewer #1: (No Response)

Reviewer #2: All comments have been addressed

2. Is the manuscript technically sound, and do the data support the conclusions?

Reviewer #1: Partly

Reviewer #2: Yes

3. Has the statistical analysis been performed appropriately and rigorously? 

Reviewer #1: Yes

Reviewer #2: Yes

4. Have the authors made all data underlying the findings in their manuscript fully available?

Reviewer #1: Yes

Reviewer #2: Yes

5. Is the manuscript presented in an intelligible fashion and written in standard English?

Reviewer #1: Yes

Reviewer #2: Yes

6. Review Comments to the Author

Reviewer #1: Thank you for greatly improving the manuscript. It is much improved and much easier to read now.

I only have generally minor comments at this point.

Line 52 - not registered - but 1.6m are estimated

62 - spacing issue

88 - strange to say approx and then sch a specific number 1442... maybe better to say estimated? or registered? also a spacing issue

89 - can you define what TB/HIV services are? treatment provision? diagnosis? referral?

Lines 114, 121 and 122 - I had a hard time understanding why you say estimate of 24% from the literature and then use much higher estimates of 69 and 57% with other refs - but that cannot all be true since 24% overall is lower than both the others.

157-8 - please cite the literature you mention.

190 is the committee really just School Of Medicine? Not Kampala or something else?

198 IQR is not correct/incomplete)

201 and 203 - no need to have separate paragraphs here with one sentence each.

203 - very specific to say approximately with 16.8% - Overall?

379 - representative for Kampala.

Table 1 - I think it would be good to describe a little more how many PP actually saw a person with presumptive TB - and document that - as this is important when you are making a case this is a population that can add a lot of TB detection and diagnosis to the NTP. It seems that at least 25% didnt see any which leads me to my only bigger point for the discussion.

The discussion is still much too long and should be reduced substantially - it is more than 5 pages now and this should be reduced to 3 pages highlighting the important parts of the findings and contrasting them with other work. It is not right to link poor practices in Kampala to not being able to meet global TB targets. The link is not that strong, and it seems a stretch to make such a connection. TB efforts can be strengthened and more poeple will be linked to care - earlier in their disease progression, but jumping from PP practice in Kampala to ending TB is too far. Some of the language in the discussion is a bit colloquial as well, and you may want to revisit.

Reviewer #2: Dear Author, I guess all the scientific concerns are addressed. However, there are few formatting issues in the clean version which are to be taken care of.

Best regards,

7. PLOS authors have the option to publish the peer review history of their article (what does this mean?). If published, this will include your full peer review and any attached files.

Reviewer #1: No

Reviewer #2: **Yes: **Sharath Burugina Nagaraja

---

## [Author Response · Author response to Decision Letter 1]

2 Nov 2023

REBUTTAL LETTER RESPONDING TO COMMENTS RAISED BY PLOS ONE EDITORIAL TEAM.

23rd October, 2023

To

Lara Vojnov

Academic Editor

PLOS ONE

Dear Editor,

Reference: PONE-D-22-31914R1

We thank you for reviewing and inviting us to submit a revised version of our research manuscript titled, ‘Private Practitioners’ practices for tuberculosis management in a city largely served by the private health sector in Uganda’. 

We have worked on the comments and made all the effort to ensure that the format of the text and references are standard.

Kindly receive our rebuttal letter responding to each of the comments raised by the reviewers. We have submitted the revised manuscript with track changes as well as a clean copy.

Responses to comments from Reviewer #1

Thank you for encouraging us to improve our manuscript and for taking time to review and make it better.

Comment #1: Line 52 - not registered - but 1.6m are estimated

Response: We changed the wording from registered to estimated.

Comment #2: 62 - spacing issue

Response: We have formatted the spacing.

Comment #3: 88 - strange to say approx and then sch a specific number 1442... maybe better to say estimated? or registered? also a spacing issue

Response: We have formatted the wording and the spacing.

Comment #4: 89 - can you define what TB/HIV services are? treatment provision? diagnosis? referral?

Response: TB/HIV services here refer to provision of both diagnostic and treatment services for TB and HIV patients.

Comment #5: Lines 114, 121 and 122 - I had a hard time understanding why you say estimate of 24% from the literature and then use much higher estimates of 69 and 57% with other refs - but that cannot all be true since 24% overall is lower than both the others.

Response: We used the 24% from a study done in Uganda many years back to calculate the sample size using Kish and Leslie formula. The study used the sample size of 630 which we got using this proportion of 24% and catering for non-response. The other proportions of 68% and 57% from a different study were used to calculate the sample size formula for comparing two proportions. But we never considered this sample size since it was smaller. We have removed this from the text since it creates some confusion.

Comment #6: 157-8 - please cite the literature you mention.

Response: We inserted the citation.

Comment #7: 190 is the committee really just School Of Medicine? Not Kampala or something else?

Response: It is Makerere University School of Medicine Research Ethics committee. We have corrected this in the text.

Comment #8: 198 IQR is not correct/incomplete)

Response: We have corrected this one.

Comment #9: 201 and 203 - no need to have separate paragraphs here with one sentence each.

203 - very specific to say approximately with 16.8% - Overall?

Response: We have combined the sentences into one paragraph and replaced ‘approximately’ with ‘overall’.

Comment #10: 379 - representative for Kampala.

Response: Added Kampala to the sentence.

Comment #11: Table 1 - I think it would be good to describe a little more how many PP actually saw a person with presumptive TB - and document that - as this is important when you are making a case this is a population that can add a lot of TB detection and diagnosis to the NTP. It seems that at least 25% didn’t see any which leads me to my only bigger point for the discussion.

Response: Thank you for this important insight into the data, indeed it raises a major point of discussion. We have added a narrative to table 1 to include this information.

Comment #12: The discussion is still much too long and should be reduced substantially - it is more than 5 pages now and this should be reduced to 3 pages highlighting the important parts of the findings and contrasting them with other work. It is not right to link poor practices in Kampala to not being able to meet global TB targets. The link is not that strong, and it seems a stretch to make such a connection. TB efforts can be strengthened and more people will be linked to care - earlier in their disease progression, but jumping from PP practice in Kampala to ending TB is too far. Some of the language in the discussion is a bit colloquial as well, and you may want to revisit.

Response: We have reduced the discussion and revised the statements as well as the language.

Response to Reviewer #2

Thank you for taking time to read through our work and recognizing the efforts put in to make it better. We have read through the whole document and worked on the formatting.

Kind regards,

Judith Amutuhaire-Ssemasaazi (Corresponding author)

---

## [Editor Report · Decision Letter 2]

14 Dec 2023

Private Practitioners’ practices for tuberculosis management in a city largely served by the private health sector in Uganda.

PONE-D-22-31914R2

Dear Dr. Amutuhaire Ssemasazi,

We’re pleased to inform you that your manuscript has been judged scientifically suitable for publication and will be formally accepted for publication once it meets all outstanding technical requirements.

Kind regards,

Stephen Michael Graham, FRACP, PhD

Academic Editor

PLOS ONE
---

## [Editor Report · Acceptance letter]

14 Jan 2024

PONE-D-22-31914R2 

PLOS ONE

Dear Dr. Ssemasaazi, 

I'm pleased to inform you that your manuscript has been deemed suitable for publication in PLOS ONE. Congratulations! Your manuscript is now being handed over to our production team.

Kind regards, 

on behalf of

Dr. Stephen Michael Graham 

Academic Editor

PLOS ONE